# EXCITATION DROPOUT: ENCOURAGING PLASTICITY IN DEEP NEURAL NETWORKS

## ABSTRACT

We propose a guided dropout regularizer for deep networks based on the evidence of a network prediction: the firing of neurons in specific paths. In this work, we utilize the evidence at each neuron to determine the probability of dropout, rather than dropping out neurons uniformly at random as in standard dropout. In essence, we dropout with higher probability those neurons which contribute more to decision making at training time. This approach penalizes high saliency neurons that are most relevant for model prediction, *i.e.* those having stronger evidence. By dropping such high-saliency neurons, the network is forced to learn alternative paths in order to maintain loss minimization, resulting in a plasticity-like behavior, a characteristic of human brains too. We demonstrate better generalization ability, an increased utilization of network neurons, and a higher resilience to network compression using several metrics over four image/video recognition benchmarks.

## 1 INTRODUCTION

Dropout [Hinton et al. (2012); Srivastava et al. (2014)] is a classical regularization technique that is used in many state-of-the-art deep neural networks, typically applied to fully-connected layers. Standard Dropout selects a fraction of neurons to randomly drop out by zeroing their forward signal. In this work, we propose a scheme for biasing this selection. Our scheme utilizes the contribution of neurons to the prediction made by the network at a certain training iteration stage.

Dropout can be interpreted as model averaging technique that avoids overfitting on training data, allowing for better generalization on unseen test data. A recent variant of dropout that targets improved generalization ability is Curriculum Dropout [Morerio et al. (2017)]. It targets adjusting the dropout rate by exponentially increasing the unit suppression rate during training, answering the question *How many neurons to drop out over time?* Like Standard Dropout [Hinton et al. (2012); Srivastava et al. (2014)], Curriculum Dropout selects the neurons to be dropped randomly. In this work, however, we target at determining how the dropped neurons are selected, answering the question *Which neurons to drop out?*

Our approach is inspired by brain plasticity [Hebb (2005); Song et al. (2000); Mittal et al. (2018); Miconi et al. (2018)]. We deliberately, and temporarily, paralyze/injure neurons to enforce learning alternative paths in a deep network. At training time, neurons that are more relevant to the current prediction are given a higher dropout probability. The relevance of a neuron for making a certain prediction is quantified using Excitation Backprop, a top-down saliency approach proposed by Zhang et al. (2016). Excitation Backprop conveniently yields a probability distribution at each layer that reflects neuron saliency, or neuron contribution to the prediction being made. This is utilized in the pipeline of our approach, named Excitation Dropout, which is summarized in Fig. 1.

In particular, we study how this approach improves generalization through utilizing more network's neurons for image classification. We report an increased recognition rate for both CNN models that are fine-tuned and trained from scratch. This improvement is validated on four image/video recognition datasets, and ranges from 1.1% - 6.3% over state-of-the-art Curriculum Dropout.

Next, we examine the effect of our approach on network utilization. Mittal et al. (2018) and Ma et al. (2017) introduce metrics that measure network utilization. We show a consistent increased network utilization using Excitation Dropout on four image/video recognition datasets. For example,

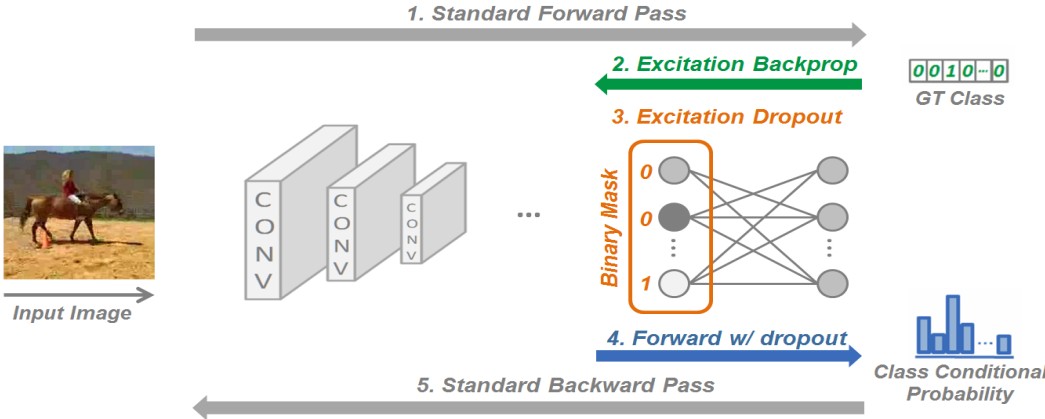

Figure 1: Training pipeline of Excitation Dropout. *Step 1:* A minibatch goes through the standard forward pass. *Step 2:* Backward EB is performed until the specified dropout layer; this gives a neuron saliency map at the dropout layer in the form of a probability distribution. *Step 3:* The probability distribution is used to generate a binary mask for each image of the batch based on a Bernoulli distribution determining whether each neuron will be dropped out or not. *Step 4:* A forward pass is performed from the specified dropout layer to the end of the network, zeroing the activations of the dropped out neurons. *Step 5:* The standard backward pass is performed to update model weights.

averaged over all four benchmarks, we get 76.55% reduction in conservative filters, filters whose parameters do not change significantly during training, as compared to Standard Dropout.

Finally, we study network resilience to neuron dropping at test time. We observe that training with Excitation Dropout leads to models that are a lot more robust when layers are shrunk/compressed by removing units. We demonstrate this when dropping the most relevant neurons, the least relevant neurons, and with a random dropping selection. This can be quite desirable for compressing/distilling [Hinton et al. (2015)] a model, *e.g.* for deployment on mobile devices.

In summary, by encouraging plasticity-like behavior, our contributions are threefold:

1. Better generalization on test data.
2. Higher utilization of network neurons.
3. Resilience to network compression.

## 2 RELATED WORK

Dropout was first introduced by Hinton et al. (2012) and Srivastava et al. (2014) as a way to prevent neural units from co-adapting too much on the training data by randomly omitting subsets of neurons at each iteration of the training phase.

Some follow-up works have explored different schemes for determining how much dropout is applied to neurons/weights. Wager et al. (2013) described the dropout mechanism in terms of an adaptive regularization, establishing a connection to the AdaGrad algorithm. Inspired by information theoretic principles, Achille & Soatto (2018) propose Information Dropout, a generalization dropout which can be automatically adapted to the data. Kingma et al. (2015) showed that a relationship between dropout and Bayesian inference can be extended when the dropout rates are directly learned from the data. Kang et al. (2017) introduces Shakeout which instead of randomly discarding units as dropout does, it randomly enhances or reverses each units contribution to the next layer. Wan et al. (2013) introduced the DropConnect framework, adding dynamic sparsity on the weights of a deep model. DropConnect generalized Standard Dropout by randomly dropping the weights rather than the neuron activations in the network. Rennie et al. (2014) proposed a time scheduling for the retaining probability for the neurons in the network. The presented adaptive regularization scheme smoothly decreased in time the number of neurons turned off during training.

Recently, Morerio et al. (2017) proposed Curriculum Dropout to adjust the dropout rate in the opposite direction, exponentially increasing unit suppression rate during training, leading to a better generalization on unseen data.

Other works focus on which neurons to drop out. Dropout is usually applied to fully-connected layers of a deep network. Conversely, Wu & Gu (2015) studied the effect of dropout in convolutional and pooling layers. The selection of neurons to drop depends on the layer where they reside. In contrast, we select neurons within a layer based on their contribution. Wang & Manning (2013) demonstrate that sampling neurons from a Gaussian approximation gave an order of magnitude speedup and more stability during training. Li et al. (2016) proposed to use multinomial sampling for dropout, *i.e.* keeping neurons according to a multinomial distribution with specific probabilities for different neurons. Ba & Frey (2013) jointly trained a binary belief network with a neural network to regularize its hidden units by selectively setting activations to zero accordingly to their magnitude. While this takes into consideration the magnitude of the forward activations, it does not take into consideration the relationship of these activations to the ground-truth. In contrast, we drop neurons based on how they contribute to a network's decision.

We compare our results against Morerio et al. (2017). To the best of our knowledge, we are the first to probabilistically select neurons to dropout based on their task-relevance.

## 3 METHOD

### 3.1 BACKGROUND

Saliency maps that quantize the importance of class-specific neurons for an input image are instrumental to our proposed scheme. Popular approaches include Class Activation Maps (CAM) [Zhou et al. (2016)], Gradient-weighted Class Activation Mapping (Grad-CAM) [Selvaraju et al. (2017)], and Excitation Backprop (EB) [Zhang et al. (2017)]. A thorough analysis of all saliency methods is out of the scope of this work, and the saliency problem in general is far from solved. We choose to use EB since it produces a valid probability distribution for each network layer. The saliency maps obtained using this approach are evaluated for spatial localization of objects and demonstrate the ability of pointing to the right region of an image [Zhang et al. (2017)].

In a standard CNN, the forward activation of neuron $a_j$ is computed by $\widehat{a}_j = \phi(\sum_i w_{ij}\widehat{a}_i + b_i)$, where $\widehat{a}_i$ is the activation coming from the previous layer, $\phi$ is a nonlinear activation function, $w_{ij}$ and $b_i$ are the weight from neuron $i$ to neuron $j$ and the added bias at layer $i$, respectively. EB devises a backpropagation formulation able to reconstruct the evidence used by a deep model to make decisions. It computes the probability of each neuron recursively using conditional probabilities $P(a_i|a_j)$ in a top-down order starting from a probability distribution over the output units, as follows:

$$P(a_i) = \sum_{a_j \in \mathcal{P}_i} P(a_i|a_j)P(a_j) \tag{1}$$

where $\mathcal{P}_i$ is the parent node set of $a_i$. EB passes top-down signals through excitatory connections having non-negative activations, excluding from the competition inhibitory ones. EB is designed with an assumption of non-negative activations. Most modern CNNs use ReLU activation functions, which satisfy this assumption. Therefore, negative weights can be assumed to not positively contribute to the final prediction. Assuming $C_j$ the child node set of $a_j$, for each $a_i \in C_j$, the conditional winning probability $P(a_i|a_j)$ is defined as

$$P(a_i|a_j) = \begin{cases} Z_j\widehat{a}_i w_{ij}, & \text{if } w_{ij} \geq 0, \\ 0, & \text{otherwise} \end{cases} \tag{2}$$

where $Z_j$ is a normalization factor such that $\sum_{a_i \in C_j} P(a_i|a_j) = 1$. Recursively propagating the top-down signal and preserving the sum of backpropagated probabilities, it is possible to highlight the salient neurons in each layer using Eqn. 1, *i.e.* neurons that mostly contribute to a specific task. We will refer to the distribution of $P(a_i)$ as $p_{EB}(a_i)$.

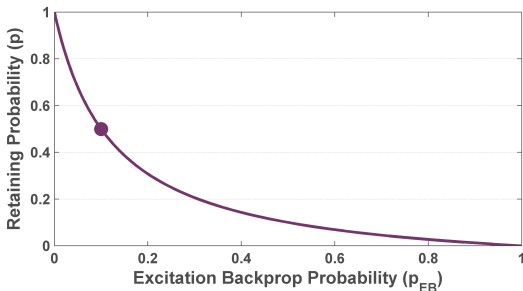

Figure 2: The retaining probability, $p$, as a function of the Excitation Backprop probability $p_{EB}$. This plot was created using $N = 10$ and a base retaining probability $P = 0.5$. In this case, when the saliency of neurons is uniform, *i.e.* $p_{EB} = 0.1$, then $p = P$ as marked in the figure.

## 3.2 EXCITATION DROPOUT

In the standard formulation of dropout [Hinton et al. (2012); Srivastava et al. (2014)], the suppression of a neuron in a given layer is modeled by a Bernoulli random variable $0 < p \leq 1$ where $p$ is defined as the probability of retaining a neuron. Given a specific layer where dropout is applied, during the training phase, each neuron is turned off with a probability $1 - p$.

We argue for a different approach that is *guided* in the way it selects neurons to be dropped. In a training iteration, certain paths have high excitation contributing to the resulting classification, while other regions of the network have low response. We encourage learning alternative paths (plasticity) through the temporary damaging of the currently highly excited path. We re-define the probability of retaining a neuron as a function of its contribution in the currently highly excited path

$$p = 1 - \frac{(1 - P) * (N - 1) * p_{EB}}{((1 - P) * N - 1) * p_{EB} + P} \tag{3}$$

where $p_{EB}$ is the probability backpropagated through the EB formulation (Eqn. 1) in layer $l$, $P$ is the *base* probability of retaining a neuron when all neurons are equally contributing to the prediction and $N$ is the number of neurons in a fully-connected layer $l$ or the number of filters in a convolutional layer $l$. The retaining probability defined in Eqn. 3 drops neurons which contribute the most to the recognition of a specific class, with higher probability. Dropping out highly relevant neurons, we retain less relevant ones and thus encourage them to awaken. We also study how this compares to dropping the least relevant neurons (Adaptive Dropout by Ba & Frey (2013)) in the Appendix.

Fig. 2 shows $p$ as a function of $p_{EB}$. To gain some intuition for Eqn. 3, we can look more closely at the graph: 1) If neuron $a_i$ has $p_{EB}(a_i) = 1$: This results in a retaining probability of $p = 0$. We do not want to keep a neuron which has a high contribution to the correct label. 2) If neuron $a_i$ has $p_{EB}(a_i) = 0$: This results in a retaining probability of $p = 1$. We want to keep a neuron which has not contributed to the correct classification of an image. 3) If neuron $a_i$ has $p_{EB}(a_i) = 1/N$, *i.e.* $p_{EB}$ is a uniform probability distribution: This results in a retaining probability $p = P$. We want to keep a neuron with *base* probability $P$ since all neurons contribute equally.

Eqn. 3 provides a dropout probability for each neuron, which is then used as the parameter of a Bernoulli distribution giving a binary dropout mask. During training, each image in a batch leads to different excitatory connections in the network and therefore has a different $p_{EB}$ distribution, consequently leading to a different dropout mask. Fig. 1 presents the pipeline of Excitation Dropout at training time, and a run-time analysis is presented in the Appendix.

## 4 EXPERIMENTS

In this section, we present how Excitation Dropout improves the generalization ability on four image/video recognition datasets in fully connected layers of different architectures. We then present an analysis of how Excitation Dropout affects the utilization of network neurons on the same datasets. Finally, we examine the resilience of a model trained using Excitation Dropout to network compression.

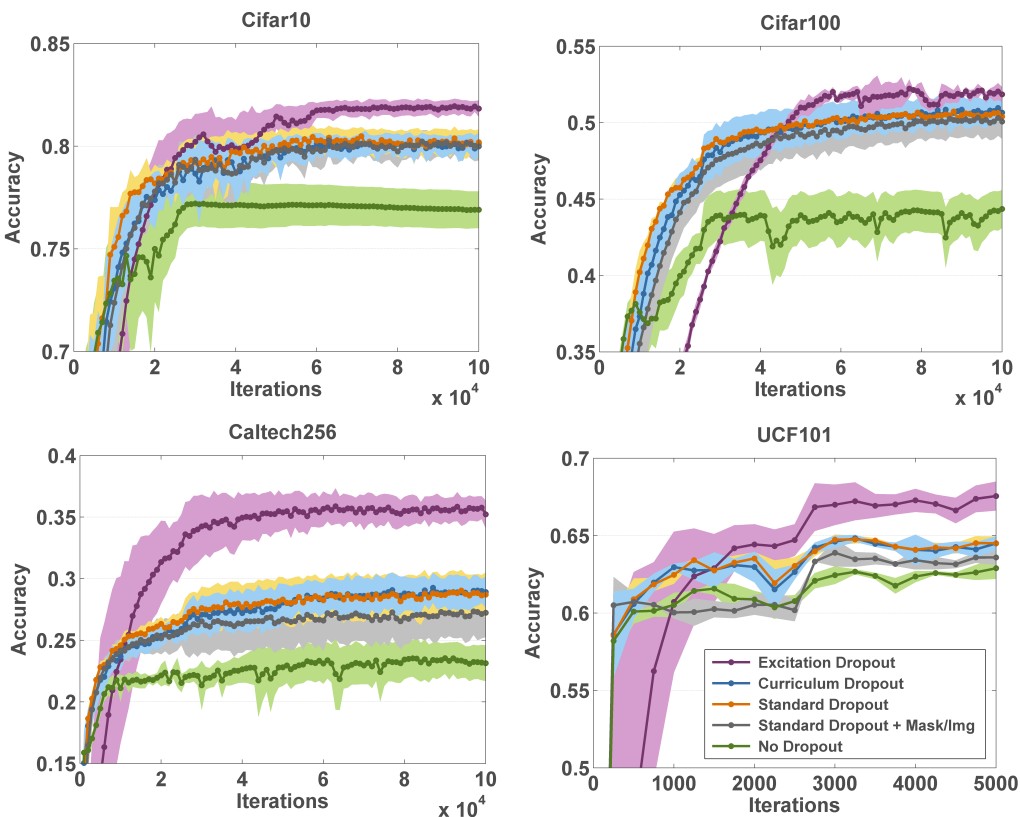

Figure 3: We compare the test accuracy of different dropout training strategies on four image/video recognition datasets: Cifar10, Cifar100, Caltech256, UCF101. Results presented here are averaged over five trained models and the standard deviation is depicted around the mean curve using a lighter shade. Excitation Dropout performs best after convergence compared to the other strategies.

## 4.1 DATASETS AND ARCHITECTURES

We present results on four image/video recognition datasets. **Cifar10** and **Cifar100** [Krizhevsky (2009)] are image recognition datasets, each consisting of $60000$ $32 \times 32$ tiny RGB natural images. Cifar10 images are distributed over 10 classes with 6000 images per class, and Cifar100 images are distributed over 100 classes with 600 images per class. Training and test splits contain $50K$ and $10K$ images, respectively. We feed the network with the original image dimensions. **Caltech256** [Griffin et al. (2007)] is an image recognition dataset consisting 31000 RGB images divided in 256 classes. We consider five different random splits of 50 train images and 20 testing images for each class. Images were reshaped to $128 \times 128$ pixel to feed the network. **UCF101** [Soomro et al. (2012)] is a video action recognition dataset based on 13320 actions belonging to 101 action classes. For this dataset we consider a frame-based action recognition task. The images are resized to $224 \times 224$ and $227 \times 227$ to fit the input layers of the VGG and AlexNet architectures, respectively.

We present results on four architectures. Relatively shallow architectures are trained from scratch, and deeper popular architectures are fine-tuned after being pre-trained on ImageNet [Deng et al. (2009)]. **Models trained from scratch:** We train the CNN-2 architecture used in Morerio et al. (2017), the state-of-the-art dropout variant, for comparison purposes. This architecture consists of three convolutional and two fully-connected layers (see Appendix). We train this network from scratch for $100K$ iterations on the datasets: Cifar-10, Cifar-100 and Caltech-256. We use mini-batches of 100 images and fix the learning rate to be $10^{-3}$, decreasing to $10^{-4}$ after $25K$ iterations. **Fine-tuned models:** We fine-tune the commonly used architectures: AlexNet [Krizhevsky et al. (2012)], VGG16 and VGG19 [Simonyan & Zisserman (2014)] pre-trained on ImageNet. We fine-tune the models for a frame by frame action recognition task on UCF101. The learning rate is fixed to $10^{-3}$ for all the processes. We fine-tune AlexNet for $5K$ while VGG16 and VGG19 for $30K$ iterations. We use a batch size of 128 and 50 images for AlexNet and VGG16/19, respectively.

| Architecture | No Dropout (%) | Standard Dropout (%) | Curriculum Dropout (%) | Excitation Dropout (%) |
|---|---|---|---|---|
| **VGG16** | 69.37 | 71.93 (+2.56%) | 72.14 (+2.77%) | **73.23 (+3.86%)** |
| **VGG19** | 71.32 | 72.52 (+1.29%) | 73.18 (+1.86%) | **74.34 (+3.02%)** |
| **AlexNet** | 62.89 | 64.50 (+1.61%) | 64.55 (+1.66%) | **67.56 (+4.67%)** |

Table 1: Test accuracy comparison between No, Standard, Curriculum and Excitation Dropout in the $fc6$ layer of three architectures: AlexNet, VGG16 and VGG19, fine-tuned for the action recognition task on UCF101. The numbers reported are the final test accuracies together with the improvements (in parenthesis) with respect to No Dropout, averaged over five trained models.

| Dataset | Metric | Standard Dropout | Curriculum Dropout | Excitation Dropout |
|---|---|---|---|---|
| *Cifar10* | # Neurons ON | 1194 ($\pm$153) | 1169 ($\pm$61) | **1325** ($\pm$61) |
| | Peak $p_{EB}$ | 0.011 ($\pm$0.004) | 0.009 ($\pm$0.001) | **0.003** ($\pm$0.0002) |
| | Entropy of Activations | 3.55 ($\pm$0.72) | 3.50 ($\pm$0.12) | **4.29** ($\pm$0.28) |
| | Entropy of $p_{EB}$ | 3.28 ($\pm$0.56) | 3.32 ($\pm$0.13) | **4.26** ($\pm$0.26) |
| | Conservative Filters$_{\Delta=0.25}$ | 1204 ($\pm$37) | 959 ($\pm$34) | **124** ($\pm$22) |
| *Cifar100* | # Neurons ON | 453 ($\pm$183) | 460 ($\pm$75) | **943** ($\pm$131) |
| | Peak $p_{EB}$ | 0.011 ($\pm$0.0004) | 0.012 ($\pm$0.0004) | **0.005** ($\pm$0.0005) |
| | Entropy of Activations | 1.67 ($\pm$0.31) | 1.70 ($\pm$0.29) | **3.21** ($\pm$0.44) |
| | Entropy of $p_{EB}$ | 1.64 ($\pm$0.27) | 1.67 ($\pm$0.26) | **3.17** ($\pm$0.41) |
| | Conservative Filters$_{\Delta=0.30}$ | 2048 ($\pm$51) | 2038 ($\pm$44) | **14** ($\pm$13) |
| *Caltech256* | # Neurons ON | 412 ($\pm$126) | 471 ($\pm$146) | **702** ($\pm$171) |
| | Peak $p_{EB}$ | 0.014 ($\pm$0.0007) | 0.013 ($\pm$0.0006) | **0.007** ($\pm$0.0003) |
| | Entropy of Activations | 1.63 ($\pm$0.32) | 1.84 ($\pm$0.35) | **2.63** ($\pm$0.23) |
| | Entropy of $p_{EB}$ | 1.58 ($\pm$0.29) | 1.77 ($\pm$0.31) | **2.59** ($\pm$0.22) |
| | Conservative Filters$_{\Delta=1.25}$ | 2048 ($\pm$46) | 2048 ($\pm$49) | **1671** ($\pm$31) |
| *UCF101* | # Neurons ON | 1120 ($\pm$25) | 1143 ($\pm$22) | **1404** ($\pm$37) |
| | Peak $p_{EB}$ | 0.007 ($\pm$0.0002) | 0.007 ($\pm$0.0002) | **0.004** ($\pm$0.0002) |
| | Entropy of Activations | 2.04 ($\pm$0.23) | 2.08 ($\pm$0.21) | **2.51** ($\pm$0.18) |
| | Entropy of $p_{EB}$ | 1.92 ($\pm$0.22) | 1.95 ($\pm$0.20) | **2.42** ($\pm$0.18) |
| | Conservative Filters$_{\Delta=0.15}$ | 3599 ($\pm$66) | 3859 ($\pm$53) | **44** ($\pm$36) |

Table 2: Different metrics to reflect the usage of network capacity in the first fully-connected layer of the CNN-2 architecture consisting of 2048 neurons and the VGG16 consisting of 4096 neurons. Results presented here are averaged over five trained models for each of the datasets: Cifar10, Cifar100, Caltech256 and UCF101 ($\sigma$ in brackets). Excitation Dropout consistently produces more neurons with non-zero activations, has a more spread saliency map leading to a lower saliency peak, has a higher entropy of both activations and saliency, and has a lower number of conservative filters; all reflecting an improved utilization of the network neurons using Excitation Dropout.

## 4.2 SETUP AND RESULTS: GENERALIZATION

In this section we compare the performance of Excitation Dropout to that of No Dropout, Standard Dropout, and Curriculum Dropout [Morerio et al. (2017)]. We train a CNN-2 model from scratch on the datasets: Cifar10, Cifar100, Caltech256. Fig. 3 depicts the test accuracies over training iterations for the three datasets averaged over five trained models. After convergence, Excitation Dropout demonstrates a significant improvement in performance compared to other methods. We hypothesize that Excitation Dropout takes longer to converge due to the additional loop (Steps 2-4 in Fig. 1) introduced in the learning process, and due to the learning of the alternative paths. We note that Excitation Dropout, during training, uses a different binary mask for each image in a minibatch, while in Standard Dropout, one random mask is employed per minibatch. To prove that it is precisely the fact that masks reflective of the particular input give rise to a boost in accuracy, and not the fact that different masks are used for different images, we add a comparison with Standard Dropout having a different random mask for each image. We refer to this accuracy as 'Standard Dropout + Mask/Img' in the plots. As expected, the latter approach is comparable to Standard Dropout in performance.

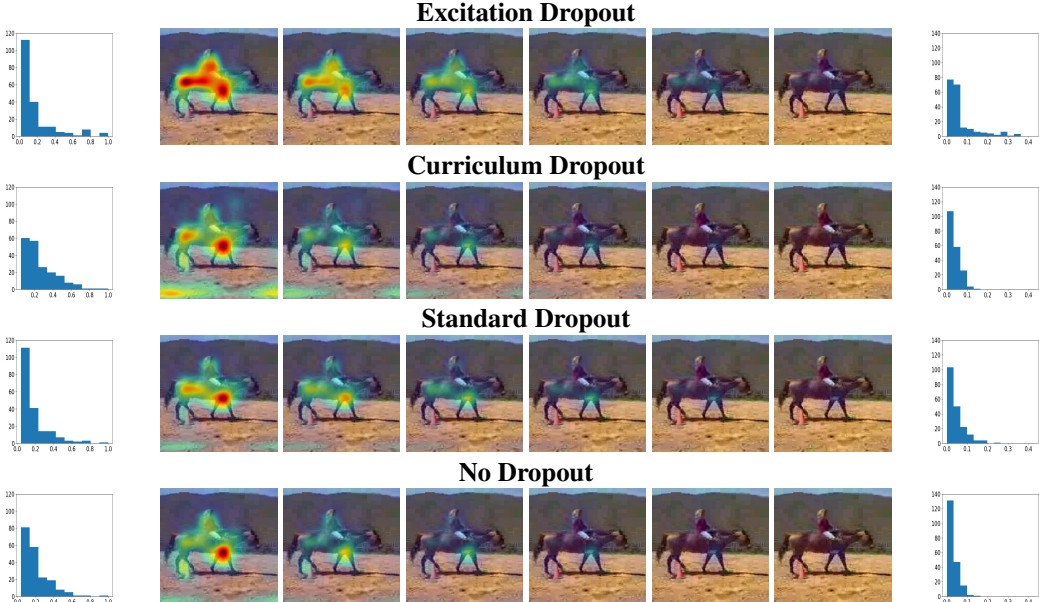

Figure 4: Visualizations for a VGG16 network fine-tuned on UCF101. The middle columns display the saliency map over the same video frame of the action *HorseRiding* while incrementally switching off the most $k$ relevant/salient neurons ($k = 0, 100, 200, \ldots, 500$) in the *fc6* layer at test time. Excitation Dropout shows more robustness when more neurons are switched off. This is demonstrated through its ability to recover more of the saliency map even when a high percentage of the most salient neurons is dropped-out. This ability reflects the alternative learnt paths. Histograms of the leftmost and rightmost saliency maps are presented to demonstrate that Excitation Dropout has a wider range of saliency values.

Next, we evaluate the effectiveness of Excitation Dropout on popular network architectures that employ dropout layers: AlexNet, VGG16, VGG19. This is done by fine-tuning on the video recognition dataset UCF101. Fig. 3 shows superior Excitation Dropout performance on AlexNet fine-tuned on UCF101. Table 1 presents more comparative results on other deep architectures by reporting the accuracy after convergence. Again, Excitation Dropout demonstrates higher generalizability on the test data for all architectures.

For fair comparison, we set $p = 0.5$ for Standard Dropout and $P = 0.5$ for the base retaining probability of Excitation Dropout in all experiments[1]. We perform dropout in the first fully-connected layer of the networks (*fc1* for CNN-2 and *fc6* for AlexNet and VGGs) for Standard, Curriculum, and Excitation Dropout. For Curriculum Dropout we fix the parameter $\gamma$ to $5*10^{-4}$ as in Morerio et al. (2017).

### 4.3 SETUP AND RESULTS: UTILIZATION OF NETWORK NEURONS

In this section we examine how Excitation Dropout expands the network's utilization of neurons through the learnt alternative paths for a certain task.

Mittal et al. (2018) introduced scoring functions to rank the filters in specific network layers including the *average percentage of zero activations*, a metric to count how many neurons have zero activations, and the *entropy of activations*, a metric to measure how much information is contained in the neurons of a layer. We analogously compute the *Neurons ON* which is the average number of non-zero activations, the *entropy of* $p_{EB}$ which is higher when the probability distribution is spread out over more neurons in a layer. We also compute the *peak* $p_{EB}$ which is expected to be lower on a more spread distribution. Moreover, Ma et al. (2017) introduced *conservative filters*: filters whose parameters do not change significantly during training. Conservative filters reduce the effective number of parameters in a CNN and may limit the CNNs modeling capacity for the target task.

---

[1]We also present a hyper-parameter sensitivity analysis for the dropout rate in the Appendix.

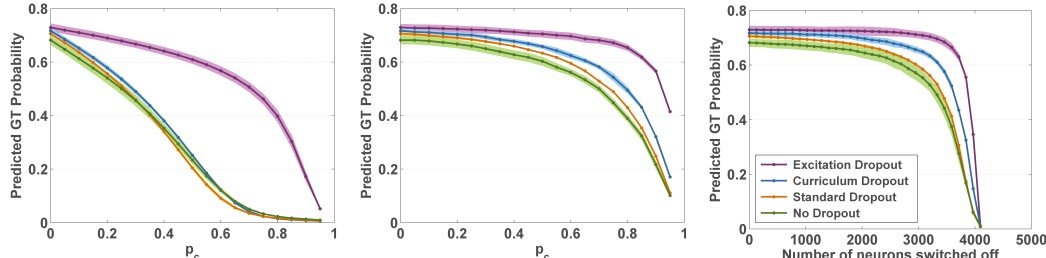

Figure 5: Robustness of predicted ground-truth class probabilities as more neurons are dropped-out for UCF101 test images. We fine-tune VGG16 with Excitation, Curriculum, Standard, and No Dropout at the $fc6$ layer, averaging results over five trained models. The standard deviation is depicted around the mean curve using a lighter shade. At test time, we switch off starting from the most relevant neurons with respect to $p_c$ (left), from the least relevant neurons with respect to $p_c$ (center), and $k$ random neurons (right). In all scenarios, Excitation Dropout shows more robustness to network compression (dropping $fc$ neurons $\equiv$ removing filters).

A conservative filter is a filter $k$ in layer $n$ whose weights have changed by $\Delta_n^k = \|\hat{w}_n^k - w_n^k\|$, where $\Delta_n^k$ is less than a threshold $\Delta$ (empirically set).

We evaluate the presented metrics for Excitation Dropout and compare against Standard and Curriculum Dropout in Table 2. This is done on the same datasets and architectures considered in Sec. 4.2. All metrics are computed for the first fully-connected layer of the CNN-2 and VGG16 nets consisting of 2048 and 4096 neurons, respectively. We compute each metric over the test set of each dataset. Excitation Dropout consistently outperforms Standard and Curriculum Dropout in all the metrics over all datasets. Excitation Dropout shows a higher number of active neurons, a higher entropy over activations, a probability distribution $p_{EB}$ that is more spread (higher entropy over $p_{EB}$) among the neurons of the layer, leading to a lower peak probability of $p_{EB}$ and therefore less specialized neurons. Averaging models having less specialized neurons results in higher robustness to information loss. We also observe a significantly smaller number of conservative filters when using Excitation Dropout. Fewer filters remain unchanged, *i.e.* do not sufficiently learn anything far from the random initialization. These results show that the models trained with Excitation Dropout were trained to be more informative, *i.e.* the contribution for the final classification task is provided by a higher number of neurons in the network, reflecting the alternative learnt paths. An analysis of such metrics over the training iterations is presented in the Appendix.

### 4.4 SETUP AND RESULTS: RESILIENCE TO COMPRESSION

In this section, we simulate 'Brain Damage' by dropping out neurons at test time. Fig. 4 demonstrates how a network utilizes the learnt alternative paths to capture the evidence of the class *HorseRiding* in a video frame of the UCF101 dataset. Given a VGG16 model fine-tuned with Excitation, Curriculum, Standard, and No Dropout at the $fc6$ layer, we show the excitation saliency map obtained at the conv5-1 layer as we drop out a fixed number of the most relevant neurons from the same layer dropout is performed upon during training. A neuron is considered to be more relevant if it has a higher $p_{EB}$. In the first column of frames of Fig. 4, the original saliency maps for the different models are shown. As already highlighted in Table 2, the original saliency map obtained from the model trained with Excitation Dropout is more spread as compared to that of the other schemes, which present more pronounced red peaks. In the following columns of Fig. 4, we present the saliency maps the model is able to restore when the $100, 200, 300, 400, 500$ most relevant neurons are dropped-out. Despite the increasing number of relevant neurons being dropped-out, Excitation Dropout is capable of restoring more of the saliency map contributing to *HorseRiding*. This means that the network with Excitation Dropout was trained to find alternative paths which belong to the same *HorseRiding*-relevant cues of the image. Despite the fact that we are considering the *worst-case* scenario, where we are switching off the *most* relevant neurons at test time, Excitation Dropout shows most robustness. More examples are presented in the Appendix.

While Fig. 4 visualizes one example qualitatively, Fig. 5 presents a complete quantitative analysis on the entire test set after training is complete. We study how the predicted ground-truth (GT) probability changes as more neurons are dropped-out at test time. On the left we present the worst case

when the neurons dropped are the most relevant to the prediction. The horizontal axis in the graph represents $p_c$, where $0 \leq p_c \leq 1$ is the cumulative sum of $p_{EB}$ of neurons which will be switched off starting from the most 'important'. The analysis is performed for $p_c = \{0, 0.05, \ldots, 0.90, 0.95\}$. In the center, we present an analogous analysis starting to drop from the 'least' relevant neurons. On the right, we present the random case (more realistic) when $k$ neurons ($k = 0, 128, 256, \ldots, 4096$) are randomly switched off. As we drop more neurons, Excitation Dropout (purple curves) is capable of maintaining a much less steep decline of GT probability, indicating more robustness against network compression. Cifar10, Cifar100, and Caltech256 show similar behavior (see analogous plots in the Appendix).

## 5 CONCLUSION

We propose a new regularization scheme that encourages the learning of alternative paths in a neural network by deliberately paralyzing high-saliency neurons that contribute more to a network's prediction during training. In experiments on four image/video recognition datasets, and on different architectures, we demonstrate that our approach yields better generalization on unseen data, higher utilization of network neurons, and higher resilience to network compression.

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

APPENDIX

**CNN-2 architecture.** Table 3 shows the details of the CNN-2 architecture adopted for Cifar10, Cifar100, and Caltech256 experiments. The size of the softmax layer depends upon the number of classes for each dataset.

| Layer Type | Layer Size | Filter Size | Padding/Stride |
|------------|------------|-------------|----------------|
| conv | 96 filters | 5x5 | 2/1 |
| max pool | | 3x3 | 0/2 |
| conv | 128 filters | 5x5 | 2/1 |
| max pool | | 3x3 | 0/2 |
| conv | 256 filters | 5x5 | 2/1 |
| max pool | | 3x3 | 0/2 |
| fc | 2048 units | | |
| fc | 2048 units | | |
| softmax | # classes | | |

Table 3: Details of the CNN-2 architecture used for experiments on the Cifar10, Cifar100, and Caltech-256 datasets.

**Least *vs.* most relevant neurons.** Popular dropout methods (*e.g.* Adaptive Dropout Ba & Frey (2013)) drop *useless* neurons with low activations during training. In this work we motivate and demonstrate that dropping neurons based on their Excitation Backprop (EB) probability has added benefits. Please note that we are not simply considering neuron activation. We demonstrate the performance of Excitation Dropout compared to the variant Adaptive Dropout [Ba & Frey (2013)] in Table 4. In essence, Adaptive and Excitation Dropout are opposites by dropping the least and most important neurons, respectively.

| Dataset | Adaptive Dropout | Excitation Dropout |
|---------|------------------|--------------------|
| Cifar10 | 76.82% | **81.94%** |
| Cifar100 | 44.55% | **52.04%** |
| Caltech256 | 23.32% | **35.77%** |
| UCF101 | 71.76% | **73.23%** |

Table 4: Accuracy comparison between Adaptive Dropout and Excitation Dropout. The numbers reported in this table are the average test set accuracy over five trained models for each dataset.

**Model complexity.** Excitation Dropout consistently outperforms Standard Dropout (SD), with zero increase in test-time computational complexity. In training, there is a moderate increase in computation: in the worst case, Excitation Dropout will take double (same O-notation complexity) the training time of SD. This will happen when the utilized Excitation Dropout maps are at the first layer of the network. If a middle layer map is used, Excitation Dropout requires an additional ***partial*** forward-backward pass. We use maps of fc layers close to the end of the network to reduce this overhead. Table 5 presents a run-time analysis for the two main architectures used in this work and compares it to that of Standard Dropout.

| | | Standard Dropout | Excitation Dropout |
|---|---|------------------|--------------------|
| **Run-time** | **CNN-2** *(1 iter)* | $0.1532 \pm 0.0064$ | $0.1885 \pm 0.0070$ (+23%) |
| | **VGG16** *(1 iter)* | $2.2928 \pm 0.0297$ | $2.8202 \pm 0.0312$ (+23%) |

Table 5: Run-time comparison: Average time of 100 iterations (in seconds, batch size=50) for a Caffe python layer on a GTX Titan X GPU and Intel(R) Xeon(R) CPU E5-2650 v3 @ 2.30GHz. In parenthesis is the percentage increase with respect to SD.

**Sensitivity analysis.** In this section we present a sensitivity analysis over the dropout rate hyperparameter. In our work, we implement Excitation Dropout with a base retaining probability $P$, and we compare that to standard dropout with a retaining probability $p$, where $P = p$. If Excitation

Dropout produced a uniform probability distribution over the desired layer, then every node would have a retain probability equal to the base probability $P$. For completeness, we add a sensitivity analysis of the parameters $p$ and $P$ in Table 6.

| Dataset | Dropout Scheme | 0.25 Dropout | 0.5 Dropout | 0.75 Dropout |
|---|---|---|---|---|
| Cifar10 | Standard | 79.16% | 80.13% | 81.19% |
|  | Excitation | **81.38%** | **81.94%** | **81.55%** |
| Cifar100 | Standard | 48.44% | 50.36% | 51.64% |
|  | Excitation | **53.23%** | **52.04%** | **51.87%** |
| Caltech256 | Standard | 26.23% | 28.73% | 32.51% |
|  | Excitation | **33.60%** | **35.77%** | **36.81%** |
| UCF101 | Standard | 71.01% | 71.93% | 72.92% |
|  | Excitation | **73.56%** | **73.23%** | **73.06%** |

Table 6: Hyper-parameter sensitivity analysis for the Standard Dropout probability, and the Excitation Dropout base dropout probability. The accuracy is reported on the test set of each dataset. The retaining probability $p$ or $P$ is one minus the dropout rate.

**Excitation Dropout in Convolutional Layers.** Excitation Dropout is a generic formulation that can be applied to any neural network layer. For a convolutional layer, a generic convolutional activation map is in the form of $[w, h, N]$, where $N$ is the number of feature maps while $w$ and $h$ are the spatial dimensions. To apply Excitation Dropout to a convolutional layer, first $p_{EB}$ is computed for each feature map $N$ as the sum of $p_{EB}$ across spatial locations $w$ and $h$. Specific 2D feature maps are then dropped-out following Eqn. 3. We test Excitation Dropout at $conv3$ of the CNN-2 architecture and obtain the following accuracy results for Cifar10: No Dropout 76.91%; Excitation Dropout at $conv3$ 78.01%; Excitation Dropout at $fc1$ 81.94%. Again, we observe an improvement respect to No Dropout, but consistent with the literature [Hinton et al. (2012); Srivastava et al. (2014)], the improvement is not as large as using dropout in fully connected layers.

**Validation on Multiple Architectures.** In this section we report additional results for Caltech256 and Cifar10 datasets considering different architectures. Table 7 reports the results of fine-tuning the deep architectures AlexNet, VGG16 and VGG19 on Caltech256 for the task of image classification. Finally, we adopt the WideResNet (WRN-28-10) architecture, a ResNet style architecture which combines batch normalization and dropout regularization techniques. WideResNet is used to obtain state-of-the-art results on Cifar10. We replace the Standard Dropout layer in the network with Excitation Dropout obtaining 3.88% test error on Cifar10 (vs. 4.17% Zagoruyko et al.s published result, BMVC16). Therefore, Excitation Dropout gives state-of-the-art result on Cifar10.

| Architecture | No Dropout (%) | Standard Dropout (%) | Curriculum Dropout (%) | Excitation Dropout (%) |
|---|---|---|---|---|
| **VGG16** | 77.97 | 79.11 (+1.14%) | 79.31 (+1.34%) | **79.92 (+1.95%)** |
| **VGG19** | 77.98 | 79.46 (+1.48%) | 79.66 (+1.68%) | **80.65 (+2.67%)** |
| **AlexNet** | 66.48 | 68.10 (+1.62%) | 68.71 (+2.23%) | **69.66 (+3.18%)** |

Table 7: Test accuracy comparison between No, Standard, Curriculum, and Excitation Dropout in the $fc6$ layer of three architectures: AlexNet, VGG16 and VGG19, fine-tuned for the image recognition task on Caltech256. The numbers reported are the final test accuracies together with the improvements (in parenthesis) with respect to No Dropout, averaged over five trained models.

**Additional Visualizations.** In this section, we present more visualizations similar to that of Fig. 4 in the main manuscript. Visualizations for a VGG16 network fine-tuned on UCF101 for the actions: *PlayingFlute*, *PlayingSitar*, and *GolfSwing* are presented in Fig.s 6,7,8, respectively. Every column displays the saliency map over the same video frame of an action while incrementally switching off the most $k$ relevant neurons ($k = 0, 100, 200, \ldots, 500$). Excitation Dropout is more robust over higher number of switched off neurons. This is demonstrated through its ability to recover the saliency map even when a high percentage of the most salient neurons is dropped-out. This is done through the alternative learnt paths, which are reflected in the higher number of non-zero activations that remain after dropout compared to other dropout strategies.

**Excitation Dropout**

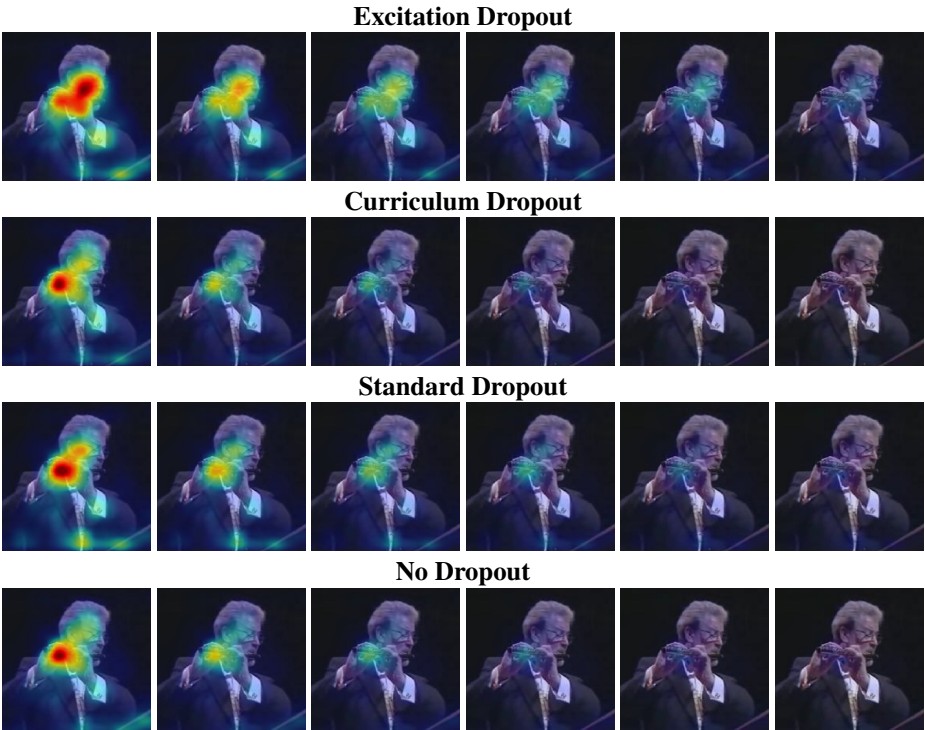

**Curriculum Dropout**

**Standard Dropout**

**No Dropout**

Figure 6: Despite the increasing number of relevant neurons being dropped out at test time, Excitation Dropout is capable of restoring more of the saliency map contributing to the specific class: *PlayingFlute*

**Additional Plots.** In this section we present the $fc$ compression results for three additional datasets: Cifar10, Cifar100, and Caltech256 (Fig. 9), using CNN-2. We study how the predicted ground-truth (GT) probability changes as more neurons are dropped-out at test time. On the left we present the worst case when the neurons dropped are the most relevant to the prediction. The horizontal axis in the graph represents $p_c$, where $0 \leq p_c \leq 1$ is the cumulative sum of $p_{EB}$ of neurons which will be switched off starting from the most 'important'. The analysis is performed for $p_c = \{0, 0.05, \dots, 0.90, 0.95\}$. In the center, we present an analogous analysis starting to drop from the 'least' relevant neurons. On the right, we present the random case (more realistic) when $k$ neurons ($k = 0, 128, 256, \dots, 2048$) are randomly switched off. As we drop more neurons, Excitation Dropout (purple curves) is capable of maintaining a much less steep decline of GT probability.

**Metric Analysis During Training.** In this section we report an extended analysis of the metrics: *# Neurons ON*, *Peak $p_{EB}$*, *Entropy of Activations*, and *Entropy of $p_{EB}$* during training. Excitation Dropout shows a higher number of active neurons, a higher entropy over activations, a probability distribution $p_{EB}$ that is more spread (higher entropy over $p_{EB}$) among the neurons of the layer, leading to a lower peak probability of $p_{EB}$ and therefore less specialized neurons. These results are observed to have consistent trends over all training iterations for all datasets considered (see Fig.s 10, 11, 12, and 13)

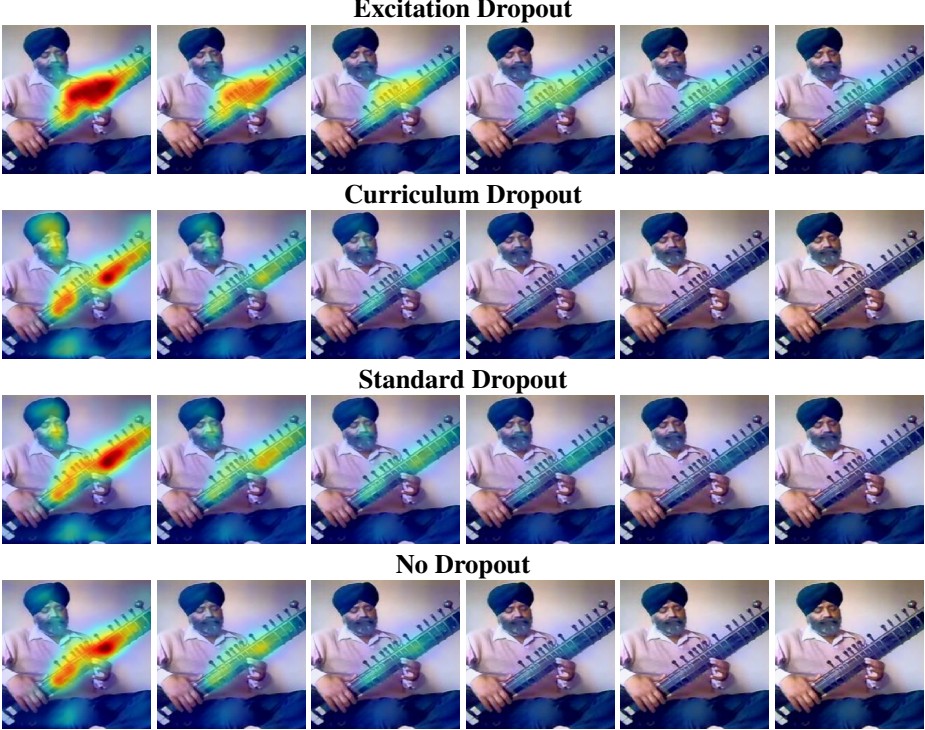

Figure 7: Despite the increasing number of relevant neurons being dropped out at test time, Excitation Dropout is capable of restoring more of the saliency map contributing to the specific class: *PlayingSitar*

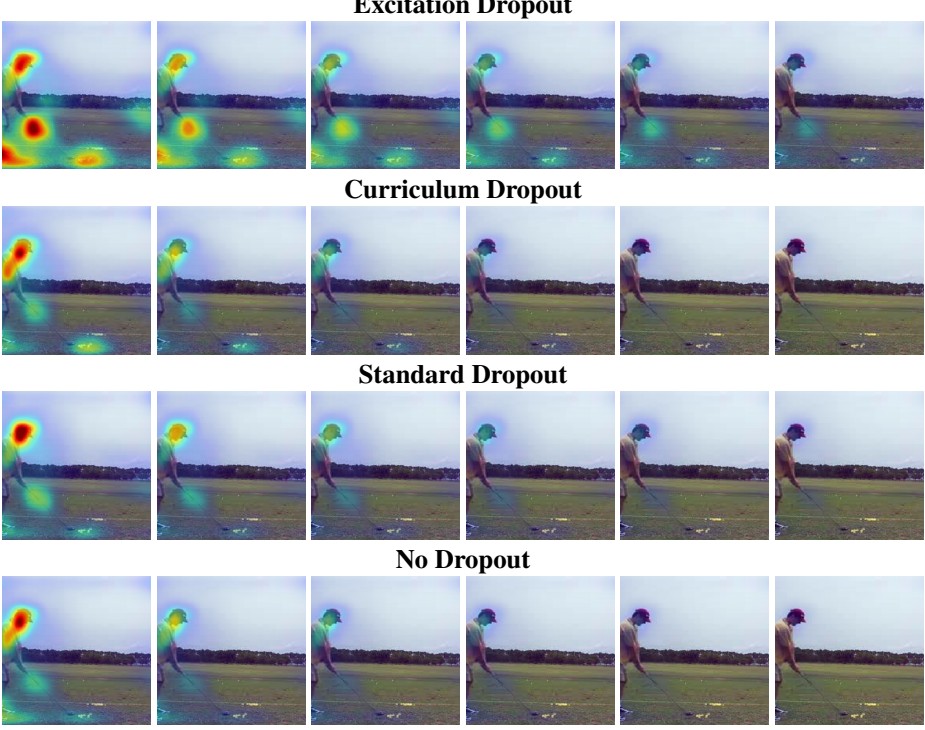

Figure 8: Despite the increasing number of relevant neurons being dropped out at test time, Excitation Dropout is capable of restoring more of the saliency map contributing to the specific class: *GolfSwing*

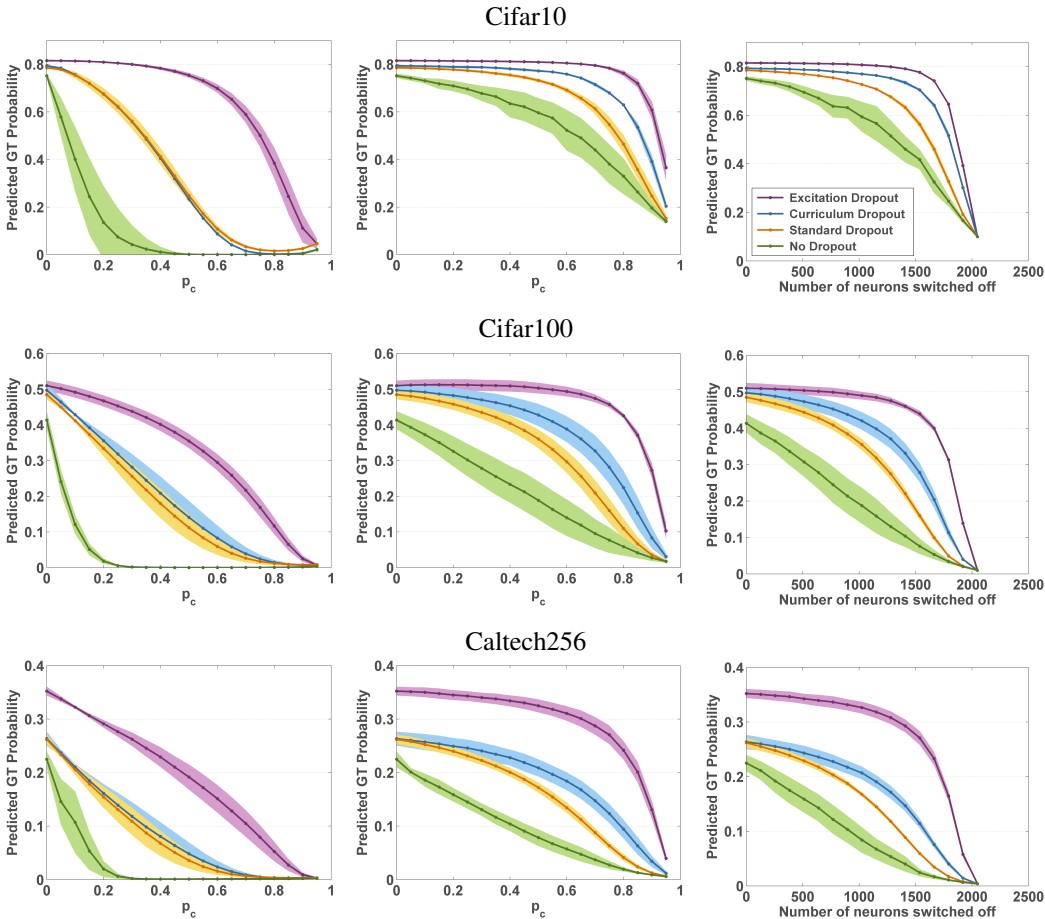

Figure 9: Robustness of predicted ground-truth class probabilities as more neurons are dropped out for each dataset's test images. We train CNN-2 from scratch with Excitation, Curriculum, Standard, and No Dropout at the $fc1$ layer, averaging results over five trained models. The standard deviation is depicted around the mean curve using a lighter shade. Left: the most relevant neurons with respect to the $p_c$ threshold are switched off. Center: the least relevant neurons with respect to the $p_c$ threshold are switched off. Right: $k$ neurons are randomly switched off. In all scenarios, Excitation Dropout shows more robustness to network compression (dropping $fc$ neurons $\equiv$ removing filters).

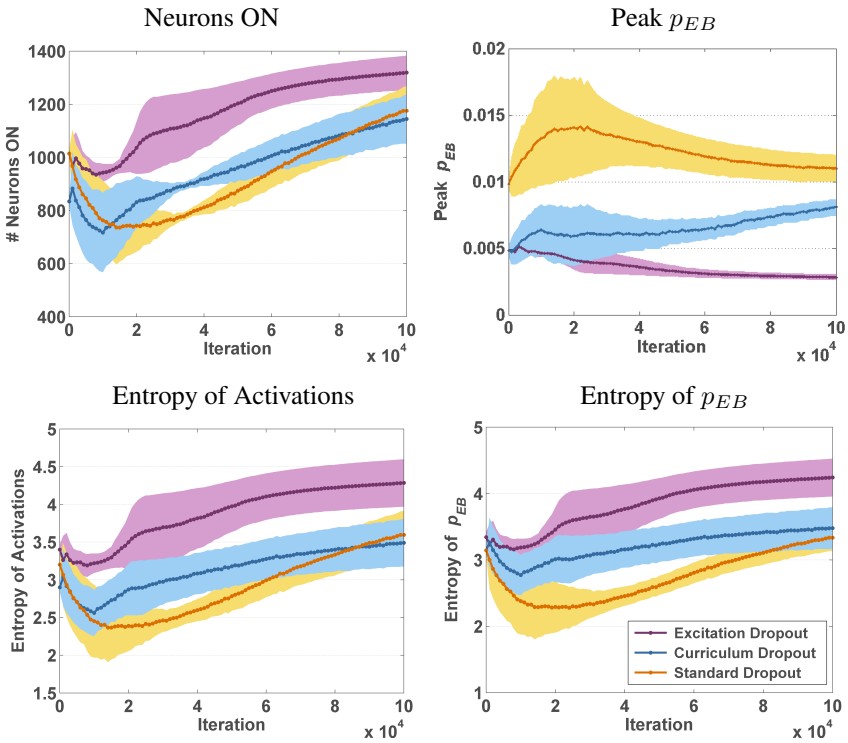

Figure 10: **Cifar10.** *# Neurons ON*, *Peak $p_{EB}$*, *Entropy of Activations*, and *Entropy of $p_{EB}$* over time during training.

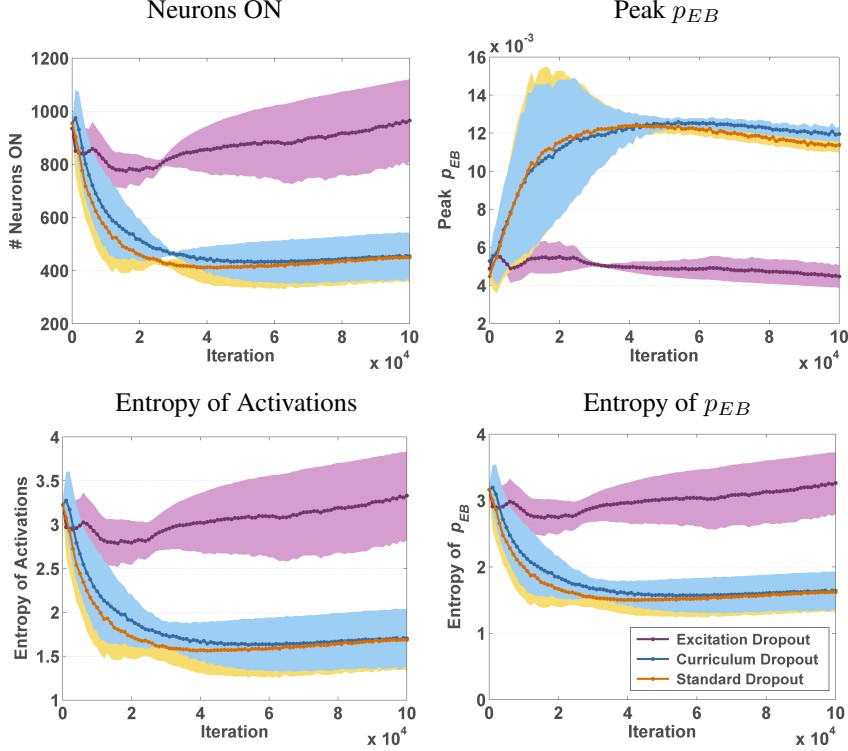

Figure 11: **Cifar100.** *# Neurons ON*, *Peak $p_{EB}$*, *Entropy of Activations*, and *Entropy of $p_{EB}$* over time during training.

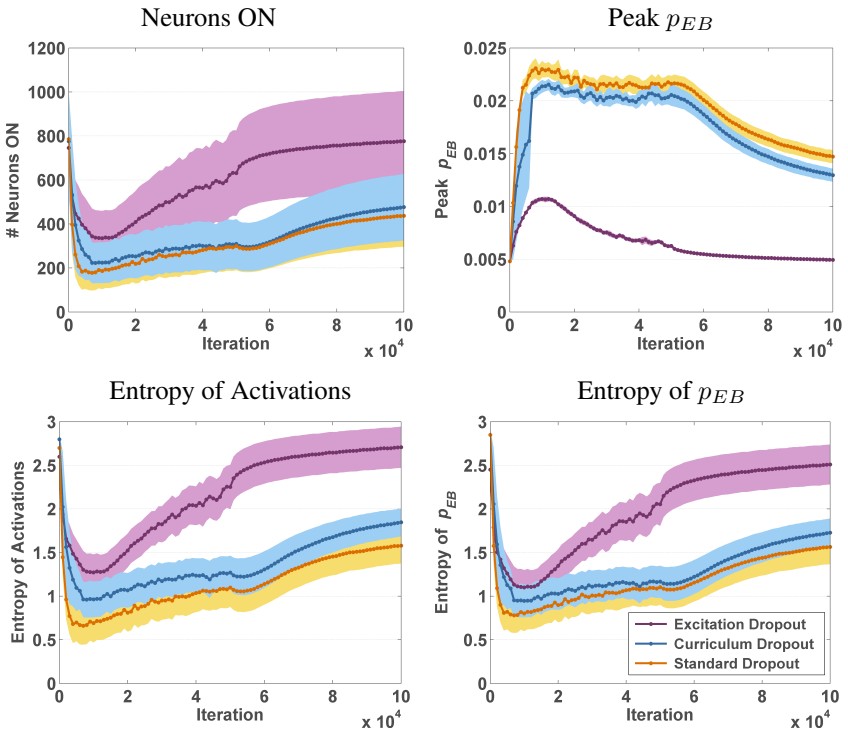

Figure 12: **Caltech256.** *# Neurons ON*, *Peak $p_{EB}$*, *Entropy of Activations*, and *Entropy of $p_{EB}$* over time during training.

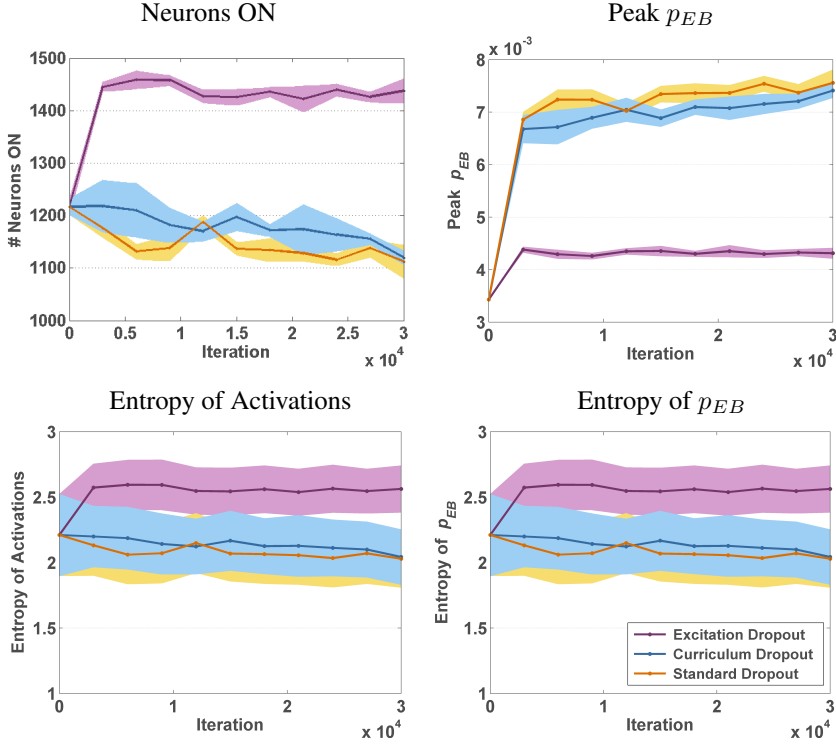

Figure 13: **UCF101.** *# Neurons ON*, *Peak $p_{EB}$*, *Entropy of Activations*, and *Entropy of $p_{EB}$* over time during training.

