# OpenReview forum: "Excitation Dropout: Encouraging Plasticity in Deep Neural Networks"
_ICLR.cc/2019/Conference_

### Official Review · AnonReviewer3 · 2018-11-02
**Alternative to Dropout**

**Rating:** 5
**Confidence:** 4

**Review:**

The authors propose a data-dependent dropout variant that produces dropout candidates based on their predictive saliency / relevance. Results are reported for 4 datasets (Cifar10, Cifar100, Caltech256 and UCF101) and 4 different models (CNN-2, AlexNet, VGG16 and VGG19), and suggest an increase in generalization performance over other dropout approaches (curriculum dropout, standard dropout, no dropout), as well as increase in the network's plasticity as measured by some existing metrics from the literature. The authors conclude that Excitation Dropout results in better network utilization and offers advantages for network compression (in the sense of neuron pruning).

Overall I find the idea to be interesting and fairly novel, and commend the authors for the fluid writing style. However, I find key issues with the testing and experiments. Specifically, the lack of confidence bounds for individual results makes it impossible to determine whether the reported incremental improvements are actually significant over those of existing approaches. Likewise, I criticize the choice of methods the authors have chosen to compare against, as several other data-dependent dropout approaches (e.g. Information Dropout) exist that may be conceptually closer (and therefore more comparable) to the proposed approach. I also question the choice of tested network architectures and the placement of the dropout layer.

The paper could be of high significance if all claims in the paper could be backed up by experiments that show the advantage of Excitation Dropout to be not a random effect. I will therefore give a lower score for the paper in its current form, but am willing to revise my rating the major points below are properly addressed.

Pros:
+ novel mechanism to improve dropout, results seemingly superior over other methods
+ achieves better utilization of network resources and achieves robustness to dropping out neurons at test time

Cons:
- results without error bars, unclear if advantage is significant
- did not compare against most relevant competing methods


MAJOR POINTS
Section 2 - The comparison to Moreiro et al. is not entirely clear. A fairer comparison would be with some of the other methods listed which also focus on answering the question of which neurons to dropout, or approaches which determine the dropout policy based on information gained from the data, such as Information Dropout (Achille & Soatto). The authors state that Morerio et al are the state-of-the-art in dropout techniques, however based on the results presented here (Figure 3) it seems to perform just as well as standard dropout. Perhaps there are architecture-specific or data-specific issues? In any case this example undermines the confidence of the claims.

Section 3.2, equation 3 - is there some theoretical underpinning as to how this equation was modelled, or was it chosen simply because it covers the expected corner cases described in paragraph 4 of this section? Also, given the intuition in this paragraph (e.g. p_EB = 1 / N), it is correct to assume this equation models the dropout probability but only for fully connected layers? What about dropout in convolutional layers? Though some previous statements do point to the usage of dropout predominantly for fully connected layers, I feel that this context is missing here and should be explicitly addressed. The caption to e.g. Table 1 seems to imply the authors add a single dropout layer in one of the fully connected layers, however this begs the question as to why this positioning was chosen - why only one dropout layer, and why precisely at that location? The scope of the claims should be adapted accordingly.

Section 4.2 - "After convergence, ED demonstrates a significant improvement in performance compared to other methods". If five trained models were used, then some sense of measure of uncertainty should be given throughout. For example, in the Cifar10 results for Figure 3, it is difficult to say whether the marginal improvement from about 80% (standard dropout and curriculum dropout) to about 82% (excitation dropout) is significant or not. Perhaps this would be less of an issue if the authors had worked with e.g. ImageNet, but for these smaller datasets it would definitely be worth to be on the safe side. I highly suspect that statistically speaking (perhaps with the exception of the results on Caltech256), the effects of all of these dropout variants are indistinguishable from each other. I urge the authors to include a measure of the standard deviation / 95% confidence interval across the models that were tested.

The results presented sub-section 4.3 do not justify the claim that the models trained with Excitation Dropout tend to be more informative. Perhaps the definition of "informative" should be expanded upon in length. Can the authors show that the alternative paths learned by the models augmented with Excitation Dropout indeed carry complimentary information and not just redundant information?

Figure 5 shows interesting results, but once again begs the question of whether there is any significant difference between standard dropout and curriculum dropout. I encourage the authors to include confidence bounds for each trace. Likewise, there is an inherent bias in the results, in that the leftmost figure compares EB and CD in the context in which EB was trained, i.e. dropping of "most salient" neurons. The comparison is one-sided, however, as no results are reported from the context in which CD was trained, i.e. dropping neurons more frequently as training progresses. Comparing these results would bring to light whether the performance boost see in Figure 5 is a function of "overfitting" to the training manner or not.
Also, I believe the results for the second column (dropping out least relevant neurons) are misleading. To the best of my understanding, as p_c increases, at some point neurons start to be dropped that actually have high relevance. This could explain why all curves start out similarly and EB slowly begins to stick out - at this point the EB models once again start to be used in the context within which they were trained, in contrast to the other approaches. The authors should perhaps also explicity clarify why this second column gives any more information than the first.



MINOR POINTS

The authors propose Excitation Dropout as a guided regularization technique. Batch normalization is another standard regularization technique which is often compared with dropout. In particular, for deep CNNs, batch normalization is known to work very well, often better than the standard dropout. In the experiments here, to what extent was batch normalization and / or any other widely utilized network regularizers used? Is it possible that the regularizing effect found here actually comes from one of these? I.e. were the models that were not trained from scratch trained with batch normalization? It would be good if more data could be provided for EB vs. other regularizing techniques, if the claim is that EB is a novel regularizer.

Section 3.1 - "We choose to use EB since it produces a valid probability distribution for each network layer". Though this is a nice property, were there any other considerations for choosing the saliency method? Recent work (Adebayo et al, "Sanity Checks for Saliency Maps") has shown that even some well established saliency techniques are actually independent from both the model and data. As this approach relies heavily on the correctness of EB, I feel that a further justification should be given to validate its use for this scenario other than just based on the type of output it produces.

Section 3.1, equation 2 - more detail and reasoning should be given as to why connections with negative weights are excluded from the computation of the conditional probability, if possible without referring the reader to the EB paper. Why is this justified? Is this probability modelled for a specific activation function?

The authors do not provide the details of the CNN-2 architecture (even in the appendix) and simply refer to another article. If the majority of the results presented in the paper are based on this network (including a reference made to a specific layer of the network in subsection 4.2) – which is not commonly known – why not to detail the network architecture and save additional effort for the reader?

How are the class-wise training and test images chosen for Caltech256 dataset?

The authors test the CNN-2 architecture on Cifar10 and Cifar100, and AlexNet, VGG16, and VGG19 on UCF101. I feel that at least a couple architectures should be validated with more than a single dataset, or the authors should justify the current matching between architectures and datasets. Table 2 is unclear regarding what models were used for what datasets (caption could be interpreted to mean that VGG16 was also used for Cifar and Caltech, however other statements seem to say otherwise).

"To prove that the actual boost in accuracy with ED is not provided by the choice of specific masks,..." I suggest that the authors rephrase or explain this sentence in more detail. To the best of my understanding, it is precisely the fact that different masks are used, each reflective of the particular input used to generate the forward activations, that gives boost in performance over "standard" dropout methods by identifying salient paths in the network.

Although it is a very important experimental detail, only in the end of sub-section 4.2, it becomes clear in which layers Excitation Dropout was applied.

Y-axis labels are missing for the left panels in Figure 3.

The authors randomly choose to abbreviate Excitation Dropout as ED in some paragraphs, while write the full form in others.

Table 2 - It is not clear that the "Neurons ON" metric refers to the "average percentage of zero activations" explained below.

Table 2 - How is peak p_EB measured? Is this an average over a set of test images after convergence? If so, I similarly suggest for confidence bounds to be introduced. It would be interesting to compare this to intermediate values (e.g. after every epoch) during training. Same question for entropy of activations and entropy of pEB. This information would be useful for reproducibility.

Table 2 - Where do the delta values in Table 2 come from? If empirically determined, it should be stated explicitly.

Table 2 - In general, because the metrics provided in Table 2 are averages (second paragraph of this Section 4.3), both (to the best of my understanding) across input subsets (e.g. averging results over many test inputs) and models (caption to Table 1), I feel Table 2 in its current form raises confusion given the lack of confidence bounds. I recommend the authors to clarify what type of averaging was done and to introduce e.g. standard deviations across all reported scores. The authors should refrain from using the term "significantly" while describing results if no statistical testing was done, or explicitly clarify their usage of this term.

Table 2 - In general, Table 2 reports results on selected metrics which, if the authors' hypothesis is correct, should have a clear trend as training progresses. An interesting idea to explore would be to include an analysis (in the appendix) of how these factors change over the course of the training procedure. Intuitively, it seems plasticity is something that should be learned slowly over time, so if these plots were to reveal something different, it would be indicative that something else is going on.

Figure 4 - Judging heatmaps is difficult as it depends on the visual perception of the reader. Thus, it is difficult to judge whether, as the authors claim, ED is indeed less "peaky" than the other alternatives. I suggest that the authors use a perceptually uniform heatmap, and to acompany these figures with e.g. the histogram of the heatmap values. Likewise, it is unclear how the multi-model aspect of the testing plays a role in generating these results. From the 5 originally trained models, how was the model selected that generated these results? Was there averaging of any kind?

Figure 5: the text is too small to be readble

Is "re-wiring" the most appropriate term to use to describe what is happening at inference time? Although different paths may be used, the network connections themselves are fixed and thus this is a potential source for confusion.

What do numbers in Table-4 in the appendix represent? Test accuracy?

---

> ### Author Response · Authors · 2018-11-22
> **Reply to AnonReviewer3 (Major Points)**
>
>
>
> Major Points
>
> 1. Curriculum Dropout (ICCV’17) reports results on a large subset of the datasets on which we report results and have publicly available code. Curriculum Dropout (ICCV’17), on average, performs better than Standard Dropout in all our experiments reported in Figure 3 and Table 1.
>
> Information Dropout only reports results on MNIST and Cifar10. The best error rate obtained for Cifar10 by Information Dropout is ~8%. We gladly provide Excitation Dropout performance results for WideResNet (WRN-28-10) on Cifar10: 3.88% test error (vs. 4.17% Zagoruyko et al.’s published result, BMVC’16). Therefore, Excitation Dropout gives state-of-the-art result on Cifar10. This is now reported in the Appendix (section titled: Validation on Multiple Architectures).
>
> Closest methods that focus on answering the question of which neurons to drop out are Adaptive Dropout (NIPS’13) and Information Dropout (TPAMI’18), both having no publicly available implementation. We chose to implement Adaptive Dropout because it has a directly comparable neuron selection strategy. Comparison against Adaptive Dropout was included in the Appendix (section titled: Least vs. most relevant neurons) of our original submission, and we refer to it in the middle of page 4 of our original manuscript. In that section we discuss how Adaptive Dropout and Excitation Dropout use opposite strategies of neuron selection: Adaptive Dropout drops with a higher probability the least active neurons, whereas Excitation Dropout drops with higher probability the most relevant neurons. We also note that Adaptive Dropout was originally introduced for an autoencoder architecture.
>
> 2. Equation 3 was designed to fit the three constraints (corner cases) described in the paper, starting from a general hyperbolic function p=(a-a*x)/(a+x). Other simpler variations such as p=1-p_EB were studied, but did not perform as well as Equation 3. The latter linear scheme shows its limitations when p_EB=1/N; the probability of retaining a neuron p becomes almost 1 (i.e. no dropout), for typical values of N (e.g. 2048,4096).
>
> Excitation Dropout can be applied to any neural network layer. We add the following clarification to Section 3.2 “N is the number of neurons in a fully-connected layer l or the number of filters in a convolutional layer l.” In response to the reviewer request, we implement Excitation Dropout (ED) in convolutional layers of the CNN-2 architecture. We apply ED to a generic convolutional activation map [w,h,N] by first calculating p_EB for each of the N feature maps as the sum of p_EB across the spatial locations (w and h). We then drop a 2D feature map as per Equation 3 and obtain the following accuracy results for Cifar10: no-dropout 76.91%, ED@conv3 78.01%, ED@fc1 81.94%. Again, we observe an improvement respect to no-dropout but, consistent with the literature [Hinton et al. (2012), Srivastava et al. (2014)], the improvement is not as large as using dropout in fully connected layers. This result is now added in the Appendix (section titled: Excitation Dropout in Convolutional Layers).
>
> 3. We now update Fig. 3 of the main manuscript to include the standard deviations for accuracies over all iterations of the five models for every dataset to statistically validate our results.
>
> 4. Entropy is a measure of average information content. In Table 2, we show that ED results in a higher entropy for neuron activations, justifying the sentence in Section 4.3, which read “These results show that the models trained with ED were trained to be more informative, i.e. the contribution for the final classification task is provided by a higher number of neurons in the network, reflecting the alternative learnt paths.” If the additional active neurons carried redundant information, we would not observe an increased accuracy.
>
> 5. We now update Fig. 5 of the main manuscript and Fig 9. of the Appendix to include the standard deviations for ground-truth probabilities of the five models for every dataset to statistically validate our results.
>
> The context in which ED is trained is: dropping the most salient neurons. The context in which CD is trained is: increasing the dropout rate over time. The context in which CD was trained cannot be replicated in testing since there are no sequential iterations at test time.
>
> We agree that the leftmost figure is obtained in the same context ED was trained, but it is added for completeness. The middle figure demonstrates the case of dropping the least salient neurons first. We note that ED in the middle figure does not exactly converge to the context in which ED was trained because by the time the most relevant neurons are dropped, the less relevant ones have been already dropped too. The rightmost figure demonstrates the case of dropping neurons selected uniformly at random (unlike the context in which ED was trained), and ED again demonstrates a more robust behavior.

---

> > ### Comment · AnonReviewer3 · 2018-11-25
> > **Reply to author response (major points)**
> >
> > I thank the authors for their exhaustive review of my comments to their manuscript. In particular I greatly appreciate their willingness to include the requested experiments and revise their existing results. I believe their attention to these points have made the manuscript considerably stronger.
> >
> > At the same time, there are 2 key open issues summarized by the first major point below. Because of these issues, I cannot recommend the manuscript for publication in its current form, and will maintain my score of 5 at confidence 4.
> >
> > Please find below my response to the authors concerning MAJOR POINTS
> >
> > 1. Once again, I do not really see a clear improvement with curriculum dropout over standard dropout in Figure 3. Moreover, the standard deviations (and only marginal improvements in average accuracy except for AlexNet) in table 1, do not justify the argument presented in the introduction that curriculum dropout is/can be considered as a state-of-the art technique.
> >
> > The Cifar-10 best error rate reported in the information dropout paper applies to the specific CNN architecture used in the experiments and should not be conveniently presented (without pointing out the differences in the network architectures) by the authors as supporting information.
> >
> > Likewise, the 4.17% test error rate on Cifar-10 using WideResNet-28-10 was obtained without any dropout. With dropout in residual blocks, the Cifar-10 test error is 3.89% (Table 6, Zagoruyko et al., BMVC’16), which is the same as achieved by excitation dropout. Therefore, the added WideResNet-28-10 experiment does not improve over the state-of-the art result. In fact, you even mention in the appendix that excitation dropout results on WideResNet-28-10 were obtained by replacing the standard dropout layers. These seem to be inconsistent results? I would have liked to see even more convicing evidence that this method should be used even with newer architectures (such as WideResNet) for which dropout is not used as frequently.
> >
> > If I have understood the response correctly, Information Dropout (ID) was not chosen for the main comparison because results for ID were only reported on MNIST and Cifar10, whereas this work is interested in a wider selection of datasets. If that is the case, this rationale should be stated to some extent in the paper. Comparison with adaptive dropout: It is interesting to compare and study model behavior when dropping out the least and most relevant neurons. However, the comparison attempt in table 4 is only based on test error. Unfortunately, adaptive dropout’s test performance is significantly lower than even standard dropout (from Figure 3 and Table 1: Cifar10 ~ 80%, Cifar 100 ~ 50%, Caltech256 ~ 28% and UCF101 ~ 71.93%). Therefore, the performance comparisons between the two techniques do not show a clear motivation and again raise the question as why a more recent method such as Information Dropout (despite the code unavailability of both adaptive and information dropout techniques) was not taken into consideration?
> >
> > In short, from these points, though the method presented here is novel, it is still difficult to see whether it is really better than competing alternatives out there.
> >
> >
> > 2. This is a good addition that makes the paper stronger. For clarity, it should be stated that Equation 3 was hand crafted given the three constraints.
> >
> > 3. Great addition.
> >
> > 5. I agree with the comment that the middle figure does not exactly converge to the context of the left figure, and that CD unfortunately is not reproducible at test time as stated in the response. That does not change however that the hidden bias in favor of ED is still there. Perhaps the claim should be reframed? What the leftmost figure shows, is that a network that is trained with ED will also be much more resistant to ED-like dropout during testing than the other approaches, but in my opinion not much else.

---

> > > ### Author Response · Authors · 2018-12-04
> > > **Round 2: Response to AnonReviewer3 (major points)**
> > >
> > >
> > >
> > > 1.
> > >
> > > (a): We now consider the same architecture (i.e. All-CNN-32) used to report results with Information Dropout (TPAMI’18) for Cifar 10 (the common dataset between the two works). We perform two experiments by replacing the two dropout layers in the All-CNN-32 architecture with 1) Excitation Dropout and 2) Curriculum Dropout. Excitation Dropout achieves ~6.27% test error while Curriculum Dropout achieves ~7.18% test error, averaged over 5 runs. The best result reported for Information Dropout (TPAMI’18) achieves ~8% test error (and so does Standard Dropout in their evaluation).  We will gladly incorporate these latest experiments in the final version of the paper and thank the reviewer for the suggested comparison. Please note that the lowest test error reported for Information Dropout (TPAMI’18) in Figure 3(b) on Cifar10 does not demonstrate an improvement over binary dropout (Standard Dropout) when considering all the convolutional filters, which is the general testing case. This is consistent with the Information Dropout paper (TPAMI’18): “Information Dropout is comparable or outperforms binary dropout, especially on smaller networks.” In our experiments, Curriculum Dropout is on average better than Standard Dropout. Please note the comparison with Curriculum Dropout is provided since it is a recently-reported method; however, regarding reviewer comments about Curriculum Dropout, we respectfully note that we are not responsible for defending the +/-  of a previously-published technique.
> > >
> > > (b): The results reported for WideResNet-28-10 architecture are consistent with the public results of Zagoruyko et al. in BMVC’16 proceedings. The error rate of 3.89% is obtained by a subsequent (later, post-BMVC) arxiv version of this work. We compare against the published BMVC paper where WideResNet-28-10 achieves 4.17% error rate without dropout and 4.39% with Standard Dropout. Excitation Dropout obtains 3.88% which is consistently lower than the public results. This is a link to the official BMVC paper: http://www.bmva.org/bmvc/2016/papers/paper087/paper087.pdf, and this is a link to an arXiv update for this paper: https://arxiv.org/pdf/1605.07146.pdf.
> > >
> > > Therefore, if the main concern is absolute performance (regardless of the model): we demonstrate results for WideResNet-28-10 using the architecture that achieves state-of-the-art results on Cifar10. If the main concern is comparing with Information Dropout (using their same architecture): we now demonstrate that Excitation Dropout outperforms Information Dropout, and that Information Dropout only does as well as Standard Dropout for the setup achieving the lowest test error. We would like to stress that Excitation Dropout is a novel idea that is validated using a significant amount of experiments and architecture/dataset comparisons compared to other work on Dropout.
> > >
> > > 2.
> > >
> > > We will gladly update the relevant sentences of Section 3.2 as follows:
> > >
> > > “Fig. 2 shows p as a function of pEB; Eqn.3 for N = 10 and P=0.5. Eqn.3 was designed to fit the following three constraints starting from a general hyperbolic function: 1)...”
> > >
> > > 5.
> > >
> > > The leftmost figure is only added for completeness. We are happy to eliminate the leftmost figure if the reviewer thinks it would create confusion to readers. The authors view the “hidden bias” in the middle figure as a feature of Excitation Dropout. The authors deem that there is no “hidden bias” in the rightmost figure where neurons are randomly dropped.
> > >
> > > Overall, our experiments provide consistent and compelling evidence that ED shows a more robust behavior to any strategy aimed at dropping out neurons, which is of course more evident in case of utilizing the same schedule used for training (leftmost plot), but it is also present in the other 2 cases, as clearly shown from the middle and rightmost plots.

---

> ### Author Response · Authors · 2018-11-22
> **Reply to AnonReviewer3 (Minor Points - part 1)**
>
>
>
> Minor Points (part 1)
>
> 1. In all experiments conducted for which results were reported in the submitted manuscript we use CNN-2, AlexNet, VGG16, and VGG19, all of which do not employ any batch normalization layers.
>
> In response to the reviewer, we conduct the following experiment on WideResNet which is used to obtain state-of-the-art results on Cifar10. WideResNet employs both batch normalization and dropout. We replace the dropout of WideResNet by our Excitation Dropout (ED) and decrease the test error from 4.17% to 3.88%.
>
> 2. The top-down signal gives a probability distribution that reflects where the model is attending. A probabilistic output is directly applicable to determining dropout probabilities.
>
> Thanks for bringing this interesting work to our attention. It is still an open problem to examine all attention methods in light of this work. However, we find it non-intuitive that EB can perform localization tasks (IJCV’17) while being independent from the model and/or data. Please note that “Sanity Checks for Saliency Maps” became publicly available Oct 28, 2018 and our manuscript was submitted Sept 27, 2018.
>
> 3. Correct, EB (IJCV’18) is designed with an assumption of non-negative activations. Most modern CNNs use ReLU activation functions, which are part of this category. Therefore, negative weights can be assumed to not positively contribute to the final prediction. In addition, using negative weights will make it difficult to normalize and obtain a probability distribution. Moreover, the saliency maps obtained using this approach are evaluated for spatial localization of objects and demonstrate the ability of pointing to the right region of an image.
>
> 4. We now update our Appendix to include the details of the CNN-2 architecture (section titled: CNN-2 architecture) and reference it in the ‘Datasets and Architectures’ section of the main manuscript.
>
> 5. We use the standard training methodology on Caltech256: 5 random splits (50 training images and 20 testing images per class). The reported result is the average of the 5 splits. We have now clarified this in Section 4.1 of the manuscript.
>
> 6. The matching used at submission time is justified as follows: smaller scale datasets were run on the shallower CNN-2, and larger scale dataset on deeper architectures such as AlexNet, VGG16, and VGG19.
>
> In response to the reviewer’s request we conduct experiments for Caltech256 on AlexNet, VGG16, and VGG19 and for Cifar10 on WideResNet. Results are consistent with our original findings. We update the Appendix to include these results in the section titled: Validation on Multiple Architectures.

---

> > ### Comment · AnonReviewer3 · 2018-11-25
> > **Response to authors (minor points)**
> >
> > Please find below my reply to the authors' response concerning MINOR POINTS (part 1 and 2):
> >
> > 2. The point is that although the top down signal does give a probability distribution, it perhaps does not necessarily follow that this is where the model is attending. In general this is the gist of the "Sanity Checks for Saliency Maps" paper, which debunks this claim for other saliency methods. Though a thorough analysis of all saliency methods of this sort is clearly out of scope of this work, it might be worth mentioning the saliency problem in general is indeed far from solved.
> >
> > 3. This short explanation would be a valuable addition to the paper.
> >
> > 6. Great addition.
> >
> > 16. Another good addition, perhaps the text should state explicitly that a random VGG16 model was selected.

---

> > > ### Author Response · Authors · 2018-12-04
> > > **Round 2: Response to AnonReviewer3 (minor points)**
> > >
> > >
> > >
> > > 2. We now mention this in Section 3.1 as follows:
> > >
> > > “Saliency maps that quantize the importance of class-specific neurons for an input image are instrumental to our proposed scheme. Popular approaches include Class Activation Maps (CAM) [Zhou et al. (2016)], Gradient-weighted Class Activation Mapping (Grad-CAM) [Selvaraju et al. (2017)], and Excitation Backprop (EB) [Zhang et al. (2017)]. A thorough analysis of all saliency methods is out of the scope of this work, and the saliency problem in general is far from solved. We choose to use EB since it produces a valid probability distribution for each network layer. The saliency maps obtained using this approach are evaluated for spatial localization of objects and demonstrate the ability of pointing to the right region of an image [Zhang et al. (2017)].’’
> > >
> > > 3. We now mention this in Section 3.1 as follows:
> > >
> > > “EB passes top-down signals through excitatory connections having non-negative activations, excluding from the competition inhibitory ones. EB is designed with an assumption of non-negative activations. Most modern CNNs use ReLU activation functions, which satisfy this assumption. Therefore, negative weights can be assumed to not positively contribute to the final prediction.”
> > >
> > > 16. We will gladly update the following sentence of Section 4.4 as follows:
> > >
> > > “Given a VGG16 model (randomly selected from the five trained models used to report results in Table1) fine-tuned with Excitation, Curriculum, Standard, and No Dropout at the fc6 layer…”

---

> ### Author Response · Authors · 2018-11-22
> **Reply to AnonReviewer3 (Minor Points - part2)**
>
>
>
> Minor Points (part 2)
>
> 7. Standard Dropout applies the same mask to all images in a batch. On the other hand, we have a different mask for every image in the batch. To demonstrate that our improvement is due to the method in which the mask is selected, and not due to having a different mask selected uniformly at random we add a baseline in which standard dropout uses a different mask for every image in a batch. The comparable result to Standard Dropout shows that the gain in accuracy is not obtained by the utilization of a different mask per sample image.
>
> The sentence has been rephrased to say: “To prove that it is precisely the fact that masks reflective of the particular input give rise to a boost in accuracy, and not the fact that different masks are used for different images, we add a comparison with Standard Dropout having a different random mask for each image.”
>
> 8. This in now stated in the first paragraph of the Experiments section.
>
> 9. Y-axis labels are now added for the left panels in Figure 3.
>
> 10. Excitation Dropout is now not abbreviated anywhere in our manuscript.
>
> 11. Correct, we now clarify this in Section 4.3 of the paper: “We analogously compute the Neurons ON which is the average number of non-zero activations, … "
>
> 12. We gladly provide standard deviations for all metrics reported in Table 2. We also add plots of all reported metrics during training in the Appendix, as requested by the reviewer (section titled: Metric Analysis During Training). The intermediate values are consistent with the final snapshot results reported in Table 2 for all the datasets, and are referenced in Section 4.3 of the main manuscript.
>
> 13. The delta values have been selected empirically following Ma et al. (PR’17). This is now explicitly stated in section 4.3 of the main manuscript.
>
> 14. We gladly provide standard deviations for all metrics reported in Table 2.
>
> 15. We gladly provide plots of all reported metrics during training in the Appendix, as requested by the reviewer (section titled: Metric Analysis During Training).
>
> 16. For Figure 4, we randomly selected one of the VGG16 models. We now add histograms to Figure 4 for k=0, 500 neurons. The histograms reflect a wider range of saliency values for Excitation Dropout with respect to all other methods.
>
> 17. The text size of Fig. 5 is now increased.
>
> 18. Agreed, the term re-wiring is not used in the manuscript anymore to avoid confusion.
>
> 19. Correct, the caption of the table has been revised to reflect that.

---

### Official Review · AnonReviewer1 · 2018-11-02
**Dropout**

**Rating:** 5
**Confidence:** 3

**Review:**

This is an interesting idea that seems to do better than regular dropout.
However, the experiment seem a bit artificial, starting with less modern network designs (VGG) that can benefit from adding dropout. State of the art computer vision networks don't seem to need dropout so much, so the impact of the paper is unclear.

Section 4.4: How does this compare to state-of-the-art network compression techniques? (Deep compression, etc)

---

> ### Author Response · Authors · 2018-11-22
> **Reply to AnonReviewer1**
>
>
>
> 1. Dropout is one of the most important regularization techniques for deep learning -adopted in WideResNet which is used to obtain state-of-the-art results on Cifar10. We gladly provide Excitation Dropout performance results for WideResNet (WRN-28-10) on Cifar10: 3.88% test error (vs. 4.17% Zagoruyko et al.’s published result, BMVC’16). Therefore, Excitation Dropout gives state-of-the-art result on Cifar10. This is now reported in the Appendix (section titled: Validation on Multiple Architectures).
>
> Some modern network architectures like ResNet and DenseNet do not have dropout layers, and rely on batch normalization for regularization. However, this does not preclude research on dropout: Ghiasi et al. (NIPS’18), Achille et al. (TPAMI’18), Cavazza et al. (AISTATS’18), Morerio et al. (ICCV’17), Kang et al. (TPAMI’17), Molchanov et al. (PMLR’17). Effectiveness of regularization methods for training neural networks is dependent on the architecture and training process, and is still an open problem (e.g. batch normalization cannot handle small batches).
>
> 2. In this work, we are not proposing a network compression technique. We propose a technique that improves network generalization on unseen data by increasing the utilization of network neurons and learning alternative paths. We then demonstrate that learning alternative paths also results in added resilience to network compression.

---

### Official Review · AnonReviewer2 · 2018-11-03
**Interesting idea but not sufficiently convincing**

**Rating:** 5
**Confidence:** 4

**Review:**

This paper presents a variation of dropout, where the proposed method drops with higher probability those neurons which contribute more to decision making at training time. This idea is evaluated on several standard datasets for image classification and action recognition.

Pros:
1. This paper has interesting idea related to dropout, and shows some benefit.
2. Paper is well-written and easy to understand.

Cons:
1. There are many variations in dropouts and they all claim superiority to others. Unfortunately, most of them are not justified properly. Excitation dropout looks interesting and has potential, but its validation is not strong enough. Use of Cifar10/100, Caltech256, and UCF 101 may be okay for concept proofing, but not be sufficient for thorough validation. Also, the reported results are far from the state-of-the-art performance of each dataset. I would recommended to add the idea to the network
 to achieve the state-of-the-art performance because it will show real extra benefit of "excitation" dropout.

2. There are many variations of dropouts including variational dropout, L0-regularization, and adaptive dropout, and the paper needs to report their accuracy in addition to curriculum dropout.

3. Dropout does not exist in many modern deep neural networks and its usability is a bit weak. It would be better to generalize this idea and make it applicable to ResNet-style networks.

4. There is no clear (theoretical) justification and intuition why excitation dropout improves performance. More ablation study with internal analysis would be helpful.

Overall, this paper has interesting idea but needs more efforts to make the idea convincing.

---

> ### Author Response · Authors · 2018-11-22
> **Reply to AnonReviewer2**
>
>
>
> 1. Adaptive Dropout (NIPS’13) reported results on MNIST and NORB, Information Dropout (TPAMI’18) and Variational Dropout (NIPS’15) reported results on MNIST and Cifar10, Curriculum Dropout (ICCV’17) reported results on MNIST, Cifar10/100, and Caltech101/256. In this work, we present results on the small scale dataset Caltech256 having ~18K images (but 256 classes), the medium scale datasets Cifar10/100 having ~60K images (10/100 classes), and on the larger scale UCF101 dataset which has ~13K videos (101 classes), from which we sample 2M frames.
>
> Dropout is adopted in WideResNet which is used to obtain state-of-the-art results on Cifar10. We gladly provide Excitation Dropout performance results for WideResNet (WRN-28-10) on Cifar10: 3.88% test error (vs. 4.17% Zagoruyko et al.’s published result, BMVC’16). Therefore, Excitation Dropout gives state-of-the-art result on Cifar10.
>
> 2. Comparison against Adaptive Dropout was included in the Appendix (section titled: Least vs. most relevant neurons) of our original submission, and we referred to it in the middle of page 4 of our original manuscript. In that section we discuss how Adaptive Dropout and Excitation Dropout use opposite strategies of neuron selection: Adaptive Dropout drops with a higher probability the least active neurons, while Excitation Dropout drops with higher probability the most relevant neurons. We note that Adaptive Dropout was originally introduced for an autoencoder architecture.
>
> Variational Dropout (NIPS’15) reports results on MNIST and Cifar10 only. The best result reported by Variational Dropout on Cifar10 has error ~23%, while Excitation Dropout with a CNN-2 architecture has error ~18% and with a WideResNet architecture has ~4% error.
>
> 3. Dropout is one of the most important regularization techniques for deep learning and it is also adopted in WideResNet. ResNet and DenseNet architectures do not have dropout layers, and rely on batch normalization for regularization. Effectiveness of regularization methods for training neural networks is dependent on the architecture and training process, and is still an open problem (e.g., batch normalization cannot handle small batches).
>
> Moreover, please note how the following dropout papers deal with computationally heavy experiments. Curriculum Dropout (ICCV’17) uses  two variants of somewhat shallow networks, LeNet and the deeper version CNN-2 we also use. Variational Dropout (NIPS’15) and Dropout (JMLR’14) use an architecture with 3 hidden layers. Adaptive Dropout (NIPS’13) uses a one-hidden-layer autoencoder. We present experiments on the medium-scale network CNN-2 for datasets trained from scratch, and on the deeper AlexNet, VGG16, and VGG19  for the larger dataset UCF101 where fine-tuning is sufficient. We also gladly provide results for Cifar10 on WideResNet and for Caltech256 on the deeper AlexNet, VGG16, and VGG19 in the Appendix (section titled: Validation on Multiple Architectures).
>
> 4. Dropout is a model averaging technique. Averaging models having less specialized neurons results in higher robustness to information loss. In Table 2 we demonstrate that Excitation Dropout has less specialized neurons (lower saliency peaks and higher entropy), as compared to Standard and Curriculum Dropout for all datasets considered (Cifar10/100, Caltech256 and UCF101). In addition, Table 2 presents an internal analysis of the network filters demonstrating that a lower number of stale/conservative filters for Excitation Dropout (as compared to Standard and Curriculum Dropout) is achieved. These considerations are now better highlighted in the last paragraph of Section 4.3.

---

### Meta-Review · Area_Chair1 · 2018-12-14
**Novel idea, but requires more convincing experiments.**

**Confidence:** 4
**Recommendation:** Reject

**Metareview:**

The reviewers overall agree that excitation dropout is a novel idea that seems to produce good empirical performance. However, they remain optimistic, but unconvinced by the experiments in their current form. The authors have done an admiral job of addressing this through more experiments, including providing error bars, however it seems as though the reviewers still require more. I would recommend creating tables of architecture x dropout technique, where dropout technique includes information dropout, adaptive dropout, curriculum dropout, and standard dropout, across several standard datasets. Alternatively, the authors could try to be more ambitious and classify Imagenet. Essentially, it seems as though the current small-scale datasets have become somewhat saturated, and therefore the bar for gauging a new method on them is higher in terms of experimental rigor. This means the best strategy is to either try more difficult benchmarks, or be extremely thorough and complete in your experiments.

Regarding the wide resnet result, while I can appreciate that the original version published with higher errors, the later draft should still be taken into account as it has a) been out for a while now and b) can been reproduced in open source implementations (e.g., https://github.com/szagoruyko/wide-residual-networks).